# Development of an Alternative Protocol to Study Muscle Fatigue

**DOI:** 10.3390/metabo15010054

**Published:** 2025-01-16

**Authors:** Daniela A. Alambarrio, Benjamin K. Morris, R. Benjamin Davis, Emily B. Grabarczyk, Kari K. Turner, John M. Gonzalez

**Affiliations:** 1Department of Animal and Dairy Science, University of Georgia, Athens, GA 30602, USA; alambarriod@uga.edu (D.A.A.); emily.grabarczyk@uga.edu (E.B.G.); kturner@uga.edu (K.K.T.); 2School of Environmental, Civil, Agricultural, and Mechanical Engineering, University of Georgia, Athens, GA 30602, USAben.davis@uga.edu (R.B.D.)

**Keywords:** electromyography, muscle contraction, muscle exhaustion, myostimulation

## Abstract

When measuring real-time in vivo muscle fatigue with electromyography (**EMG**), data collection can be compromised by premature sensor removal or environmental noise; therefore, the objective of this study was to develop a postmortem in vivo methodology to induce muscle fatigue and measure it using EMG. Barrows (*N* = 20) were stratified by weight and randomly allocated into one of two treatments. The treatments consisted of barrows being subjected to a hog electric stunner super-contraction cycle (**ES**) or not (**CON**) postmortem. The right hind limb bicep femoris (**BF**) and semitendinosus (**ST**) were selected for ambulatory movement simulation using electronic muscle stimulation (**EMS**). Muscle workload during EMS was measured with EMG using median power frequency (**MdPF**) and root mean square (**RMS**) as indicators of action potential velocity and muscle fiber recruitment. Ambulatory movement was induced and recorded for 20 min with a 4:4 duty cycle at 70 Hz. Muscle biopsies were collected pre- and post-EMS for metabolite analyses to corroborate muscle fatigue onset. There was a TRT × Muscle interaction for normalized RMS percentage (*p* < 0.01), where BF from CON barrows had greater values (*p* < 0.01). There were no interactions or TRT main effects for the MdPF normalized value (*p* ≥ 0.25), but there were Period and muscle effects on MdPF (*p* < 0.01). Bicep femoris had smaller (*p* < 0.01) MdPF than ST. The percentage of MdPF decreased (*p* < 0.01) by Period 5 compared to the other Periods, which did not differ from each other (*p* ≥ 0.38). There were TRT × Muscle and Muscle × Period interactions for ATP muscle concentration (*p* ≤ 0.03). The concentration of CON BF ATP was greater (*p* < 0.01) than that of ES BF and CON and ES ST, which did not differ from each other (*p* ≥ 0.11), but the APT concentration tended to differ between ES BF and ES ST (*p* = 0.06). Semitendinosus ATP concentration decreased (*p* < 0.01) post-EMS compared to ST pre- and BF pre- and post-EMS (*p* ≥ 0.29), but BF and ST concentration tended to differ pre-EMS (*p* = 0.07). The data indicated that EMS is a valuable tool for replicating ambulatory movement or physical activity, but super-contraction is not a means to accelerate postmortem muscle fatigue onset. Therefore, further refinement, such as longer EMS stimulation time, should be considered.

## 1. Introduction

Fatigued pig syndrome and non-ambulatory pigs are industry terms used to categorize pigs that demonstrate no signs of injury or disease but are unable to walk [1,2]. Fatigued pigs often display open-mouth breathing, skin discoloration, muscle tremors, and abnormal vocalization after transportation [1,3,4]. In the 2015 production chain, transport losses due to fatigued pigs cost the industry USD 41 million [5], representing an increase of 0.19% from Ritter et al. [2]. Utilizing data from Ritter et al. [5], Morris et al. [6] estimated a loss of 335 million 4-oz boneless meals, reducing the pork industry’s ability to contribute to food security in the U.S. After transportation, different physiological mechanisms affect fatigue onset; however, the literature has not identified a specific cause.

Enoka and Duchateau [7] stated that fatigue occurs due to one or several physiological processes responsible for generating contractile force becoming impaired. Enoka and Stewart [8] suggested that muscle fatigue could be explained by four neural and neuromuscular mechanisms. The first mechanism is dependent on the task performed, the second is dependent on the force amount exerted by the motor unit, the third is dependent on the muscle optimizing total muscle force while using minimal resources (adaptation), and the fourth is dependent on the perceived task difficulty. Because pigs are not trained, and it is impossible to determine their comprehension of task difficulty, one could hypothesize that the majority of their muscle fatigue is due to mechanisms one and two. Numerous studies demonstrated skeletal muscle performance capacity impairment, failure to maintain force or power output during muscle contraction, were related to muscle fatigue [7,9]. During activity, energy demands and limitations in adenosine triphosphate (ATP) turnover and production are common factors associated with muscle fatigue onset [10]. Other substrates related to fatigue include pyridine nucleotides, nicotinamide adenine dinucleotide (NAD+), and nicotinamide adenine dinucleotide + hydrogen (NADH) because of their role in muscle energy metabolism. When studying fatigue, the literature commonly evaluates essential metabolites levels through invasive techniques [11,12]; however, newer technologies developed in human physiology allow non-invasive muscle fatigue quantification.

Surface electromyography (EMG) is the primary method used to record muscle electrical signals in conscious animal locomotion [13]. Fatigue is reflected in EMG signals with an amplitude increase, followed by a decrease in spectral frequency [7]. In fatiguing muscle, muscle load or fiber recruitment and the velocity of motor unit action potential are estimated through root mean square (RMS) and median power frequency (MdPF), respectively [14,15,16]. Eijsbouts et al. [17] and Banerjee et al. [18] demonstrated that muscle contractions from electrical muscle stimulation (EMS) instigate muscle exhaustion. Patients with muscle mass impairment and movement restriction are treated with EMS and EMG to replicate muscle contraction and accelerate fatigue onset [19,20]. De Luca et al. (2010) denoted issues with EMG data accuracy because electrode signals are restricted by superficial musculature, adipose tissue thickness, poor electrode attachment, and environmental noise [21]. When studying fatigue onset, Alcocer [22] observed strikes on walls, penning, or equipment, and other aggressive movements were included in EMG signals during performance testing, while Hennesy et al. [23] noted that pigs had to be removed from analyses because they spent more than 25% of their time off course. Therefore, the objective of this study was to, for the first time, develop a postmortem in vivo methodology to induce muscle fatigue and measure it using EMG to eliminate EMG signal interference and improve data reliability.

## 2. Materials and Methods

Our study methods were evaluated and approved by the University of Georgia (UGA) Institutional Animal Care and Use Committee (#A2022 07-066-Y1-A0).

### 2.1. Data Collection

Barrows (*N* = 20; CG36 ’ P26, Choice Genetics, West Moines, IA, USA) were transported from the UGA Swine Center (Athens, GA, USA) to the UGA Meat Science Technology Center (Athens, GA, USA). Upon arrival, the barrows were weighed, ranked heaviest to lightest, and randomly allocated into one of two treatments (**TRT**) within each two barrow strata. The treatments consisted of the barrows being subjected to a hog electric stunner (Best and Donovan, Cincinnati, OH, USA) super-contraction cycle postmortem (**ES**) or not (**CON**). The barrows were rendered insensible to pain with a captive bolt (Jarvis Product Co., Middletown, CT, USA), and a super-contraction cycle was applied according to each TRT. The super-contraction cycle consisted of two 9 s electrical super-contractions at 570 V and 0.5–2.0 A. Following treatment application, the barrows were hung by their right leg, exsanguinated, and allowed to bleed for five minutes.

Immediately after bleeding, the right hind limb bicep femoris (**BF**) and semitendinosus (**ST**) were subjected to initial muscle biopsy approximately 2.54 cm proximal to EMG electrode location, which was positioned on each muscle’s mid-belly. A 6-gauge piercing needle (Precision Needles, Denver, PA, USA) was used to create muscle tissue access. Using Mammotome elite biopsy gun entry (Mammotome, Cincinnati, OH, USA), 40 ± 0.5 mg was collected, placed in 2 mL centrifuge tubes, submerged in liquid nitrogen for immediate freezing, and stored at −80 °C. Positive and negative EMS electrodes were place 2.5 cm proximal and distal to the muscle mid-belly using IStim Super Soft squared 2 × 2 tens (Everyway Medical Instruments Co., New Taipei City, Taiwan). A surface EMG (Noraxon USA, Scottsdale, AZ, USA) electrode was positioned between negative and positive EMS electrodes to record muscle contraction. Muscle contraction was induced with EMS iStim EV-805 4CH Digital TENS/EMS (Everyway Medical Instruments Co., New Taipei City, Taiwan) set at 70 Hz pulse frequency and 4:4 duty cycle (4 s on, 4 s off) for 20 min. Final muscle biopsies were collected directly under EMG electrode attachment following stimulation.

### 2.2. Electromyography Analyses

Using a custom program in Noraxon MR 3.16 (Noraxon USA, Scottsdale, AZ, USA), EMG data were collected for each electrical burst corresponding to a muscle contraction. Electromyography amplitude and frequency characteristics were derived as RMS and MdPF, respectively, utilizing MATLAB (MathWorks, Natick, MA, USA). The data were filtered using artifact detection/removal and thresholding algorithms on MATLAB. Electrical signals from EMS were deleted, and EMG data were reconstructed via cubic interpolation through the gap (Figure 1). Following Alcocer et al.’s (2024) methods, high and low frequencies were removed, and only absolute values were considered in the data set [22].

In Excel (Microsoft 365, Microsoft Corporation, WA, USA), the data were organized and processed for statistical analyses by muscles separately. Root mean square and MdPF data were processed using different normalization methods. The data were normalized to the first 30 s of each individual barrow stimulation, separately. Each second of stimulation was divided by the normalization value and multiplied by 100 to yield a normalization value percentage. Percentages were averaged every four minutes to yield five Period values for statistical analysis.

### 2.3. Metabolite Analyses

Frozen muscle biopsy samples were sent to the Metabolic Core Facility at the University of Iowa for an LC-MS redox panel of adenosine monophosphate (**AMP**), adenosine diphosphate (**ADP**), ATP, NAD+, NADH, guanosine diphosphate (**GDP)**, guanosine monophosphate (**GMP**), and guanosine triphosphate (**GTP**) concentrations. The samples were lyophilized, homogenized, dried, and reconstituted in acetonitrile. Data of LC-MS were acquired with a Thermo Q Exactive hybrid quadrupole Orbitrap mass spectrometer with a ZIC-pHILIC guard column. The liquid chromatograph used was a Millipore SeQuant ZIC-pHILIC, and the mass spectrometer data acquisition was performed in a range of *m*/*z* 70–1000 with resolution of 70,000. The acquired data were processed using Thermo Scientific (Waltham, MA) TradeFinder 4.1 software, and AMP, ADP, ATP, NAD+, NADH, GDP, GMP, and GTP were identified based on the University of Iowa Metabolomics Core facility standard-confirmed in-house library. The data were normalized to the sum of all of the measured metabolite ions in the sample and reported as a mass-to-charge ratio (*m*/*z*).

### 2.4. Statistical Analyses

The metabolite and EMG data were analyzed as a split-plot design with repeated measures using barrows as experimental units. Barrows served as the whole plot and muscles served as the subplot. Fixed effects included treatment, muscle, and their interaction, and the random effect was animal × treatment. The Period served as the repeated measure, the barrow within the treatment served as the subject, and compound symmetry was used as the covariance structure. All models were analyzed using the MIXED procedure of SAS 9.3 (SAS Inst., Cary, NC, USA). Pairwise comparisons between the least square means of the factor level comparisons were computed using the PDIFF option of the LSMEANS statement. Statistical significance was determined at *p* ≤ 0.05 and trends at 0.05 < *p* < 0.10.

## 3. Results

There was a TRT × Muscle interaction for normalized RMS percentage (*p* < 0.01; Figure 2A); however, there were no other interactions observed (*p* ≥ 0.67). Bicep femoris from CON barrows had greater values (*p* < 0.01) of RMS than that of BF ST and ST CON and ES, which did not differ from each other (*p* ≥ 0.11).

There were no interactions or TRT main effect for the MdPF normalized value (*p* ≥ 0.25; Figure 2B), but there were Muscle and Period main effects (*p* < 0.01). Bicep femoris had smaller values (*p* < 0.01) of MdPF than that of ST. The percentage of MdPF decreased (*p* < 0.01) by Period 5 compared to that of the other Periods, which did not differ from each other (*p* ≥ 0.38).

There was a TRT × Muscle × Period interaction for muscle concentration of AMP (*p* = 0.05; Figure 3). The concentration of AMP was greater (*p* ≤ 0.05) post-EMS in CON ST compared to that of the other AMP concentrations, which were not different from each other (*p* ≥ 0.11); however, the AMP concentration of CON ST tended to differ pre- and post-EMS (*p* = 0.07).

There were TRT × Muscle and Muscle × Period interactions for ATP muscle concentration (*p* ≤ 0.03). The concentration of CON BF ATP was greater (*p* < 0.01; Figure 4A) than that of ES BF and CON and ES ST, which did not differ from each other (*p* ≥ 0.11); however, APT concentration tended to differ between ES BF and ES ST (*p* = 0.06). Additionally, ST ATP concentration decreased (*p* < 0.01; Figure 4B) post-EMS compared to ST pre- and BF pre- and post-EMS (*p* ≥ 0.29), but BF and ST concentration tended to differ pre-EMS (*p* = 0.07).

There were no interactions or TRT and Muscle main effects for ADP concentration (*p* ≥ 0.21; Table 1), but muscle concentration tended to have greater ADP concentration post-EMS than pre-EMS (*p* = 0.06). There were no interactions or main effects for NADH concentration (*p* ≥ 0.31).

There was a tendency for TRT × Period interaction (*p* = 0.09) for NAD+ muscle concentration where there was no difference (*p* = 0.88) between the CON pig’s time periods, but there was less (*p* = 0.03) NAD+ post-EMS compared pre-EMS (Figure 5A). There was a Muscle × Period interaction (*p* = 0.03) for GDP concentration (Figure 5B) where there was no difference (*p* = 0.65) between pre- or post-BF EMS concentrations, but the ST GDP concentration was greater (*p* < 0.01) post-EMS compared to pre-EMS. There tended to be a TRT × Period interaction for GMP concentration (Figure 5C) where pre- and post-EMS concentrations did not differ (*p* < 0.59) for CON pigs, but the post-EMS concentration was greater (*p* < 0.01) than the pre-EMS concentration for the ES pigs. There was a TRT × Period interaction for GTP muscle concentration (*p* = 0.03; Figure 5D). The Control and ES GTP concentrations increased post-EMS compared to pre-EMS (*p* ≤ 0.01).

## 4. Discussion

Electromyography can be a valuable technology when measuring barrow muscle fiber activity during muscle fatigue onset [16]; however, EMG reliability and data accuracy can be compromised during in vivo data collection. De Luca et al. [21] and Williams et al. [24] observed EMG signal interference between skin electrode attachment and active muscle fibers because other muscle power or environmental noise were integrated into the data set. Similarly, Alcocer [22] reported that strikes, aggressive movements, and sensor detachment resulted in 45% of EMG signals being categorized as incomplete or flawed, which resulted in inflated variability.

Epilithic seizures and involuntary muscle contractions are caused by electrical stunning that accelerates muscle metabolism [25]. In physical therapy, EMS prevents skeletal atrophy in patients suffering from muscle mass impairment [18,19,26]. Therefore, the current study used EMS-induced super-contraction to substitute performance testing and replicate muscle contraction or ambulatory movement. In the current study, the EMS frequency was set at 70 Hz. Jones et al. [27] stated that EMS frequency greater than 50 Hz recruits fatigue fast-twitch muscle fibers (type IIA, type IIX, and type IIB) more easily. The current study did not analyze muscle fiber isoform directly under EMG and EMS electrodes; however, Hennesy et al. [23] reported that 52% of BF and ST muscle fibers in finished barrows are type IIB. Muscle load and muscle fatigue were monitored through EMG MdPF and RMS. Median power frequency was utilized to record motor unit action potential velocity moving along muscle fibers, and RMS was used to record muscle load by comparing active muscle fiber to the number utilized during rest [15,16,23]. As subjects reach muscle exhaustion, MdPF is expected to decrease as muscle contraction repetition increases [26,28], while RMS initially increases, indicating fiber recruitment, followed by a decrease representing fiber exhaustion [29,30].

In the current study, there were no Period differences in RMS that indicated muscle fatigue onset or an increase in muscle fiber recruitment. In barrows with similar performance testing, Hennesy et al. [23] and Noel et al. [16] reported that RMS increased by 40–66% by the end of performance testing, while CON BF had a 9% greater RMS in the current study. Muscle activation and fiber recruitment differences may be accredited to differences in animal physiology and postmortem data collection. Conversely, in the current study, MdPF was affected by Period, decreasing by 12% on average by the end of testing, suggesting that muscle exhaustion was reached. Cockram et al. [28] also observed decreased MdPF value in sheep ST exposed to prolonged walking; however, in barrows’ BF and ST, no Period MdPF differences were observed at the end of performance testing [23]. The absence of MdPF TRT differences in the current study indicates that super-contraction did not accelerate muscle fatigue onset or replicate performance testing.

Fatigued pig syndrome is a multifactorial process, and Hamilton et al. [31] reported physical exercise and physiological or emotional stress as factors causing fatigue. Fatigued pigs undergo metabolic acidosis indicated by the accumulation of lactic acid, decreased bicarbonate, and other blood parameters, but limitations in energy supply are the most common hypotheses behind muscle fatigue. In the current study, muscle metabolites were quantified to discover if super-contraction and EMS induce muscle fatigue. Kemmler et al. [19] and Kortianou et al. [32] observed that EMS increased energy expenditure. In the current study, super-contraction decreased the BF ATP overall content by 36%, while EMS increased ST energy consumption by 36%. Differences may be accredited to the muscles consuming energy differently according to their fiber type and fiber type response to electrical stimulation type and frequency [27]. The results can also be explained by England et al.’ [33] hypothesis, which stated that when mitochondria are electrically stimulated, muscle postmortem metabolism switches from being an ATP producer to an ATP consumer. The lack of aerobic ATP synthesis from ADP in muscle mitochondria results in anaerobic depletion, which also causes ATP disappearance [34]. In human muscle fatigue onset, ATP reductions are accompanied by metabolic by-product accumulations, such as hydrogen ions, AMP, and ADP [35]. The current study only observed AMP accumulation in CON BF post-EMS. In non-electrically stimulated porcine longissimus lumborum, Muroya et al. [36] observed that AMP increased and ADP decreased 5 h postmortem; however, in beef longissimus lumborum, England et al. [33] did not observe changes in AMP and ADP content 1 h postmortem in ES and CON muscles, but muscle fatigue was not considered in these studies.

The other metabolites considered in muscle fatigue quantification are NAD+ and NADH [37]. Duboc et al. [11] and Mayevsky and Rogatsky [38] observed increased NAD+ levels followed by NADH levels declining during muscle contraction in amphibians and rats. Jöbsis and Stainsby [39] and Wolfe et al. [40] observed the same trend in mammalian muscle. Additionally, in hypoxic conditions, NAD+ is not regenerated from NADH, which increases its content because of anaerobic glycolysis [41]. In the current study, NAD+ increased post-EMS in ES barrows, but NADH was unaffected. The current study also evaluated the biochemical changes in purine metabolism to indicate that ATP-related compounds developed postmortem. The content of GDP, GMP, and GTP increased post-EMS in the current study by 41, 71, and 41%, respectively. Muroya et al. [36] observed an accumulation of GDP and GMP, while GTP content decreased postmortem; however, the metabolites were analyzed over time while, in the current study, the metabolite profile was collected once post-EMS. The authors hypothesized that GTP was exhausted, similarly to ATP, and limited by metabolite regeneration. Similarly, Ohmura et al. [42] observed that GTP decreased 0.83-fold in thoroughbred horses post strenuous treadmill running. In the same study, the GTP content decreased 10% more than that of ATP post-exercise because GTP is reversely transferred to ATP. The results may differ from the available literature because this study is the first to attempt muscle fatigue replication and quantification post-exsanguination. Skeletal muscle still tries to achieve ATP and glycogen homeostasis balance postmortem; however, metabolites are not utilized or recycled like they are in living fatigued muscle tissue.

## 5. Conclusions

Median power frequency, ATP, and purine metabolite results indicate that EMS can be utilized to replicate physical activity or ambulatory movement early postmortem; however, other metabolite data indicated that anaerobic glycolysis was the metabolic pathway associated with the measures indicating muscle fatigue onset. The use of electrical stunning to induce super-contraction did not influence muscle fatigue onset differences; however, the lack of large differences may be due to body temperature loss associated with the postmortem condition. Conducting EMS over a longer period could possibly produce an EMG profile more consistent with aerobic muscle fatigue or adapting other technologies such as near-infrared spectroscopy to replicate the protocol could yield a better oxidative metabolism measure; however, this technique will compromise the ability to use pigs for meat and downstream applications, because it requires anesthetizing barrows.

## Figures and Tables

**Figure 1 metabolites-15-00054-f001:**
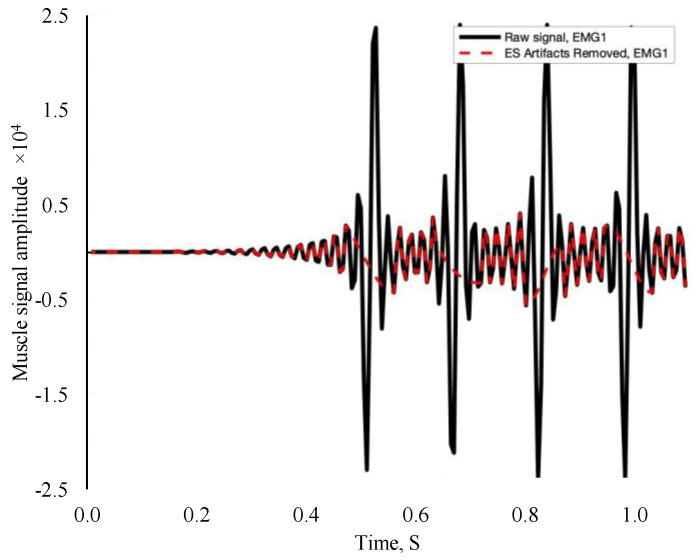
Example filtered electromyography signal. The black line represents the electrical stimulation signal, while the red dash line represents EMG data reconstruction and independent muscle activity.

**Figure 2 metabolites-15-00054-f002:**
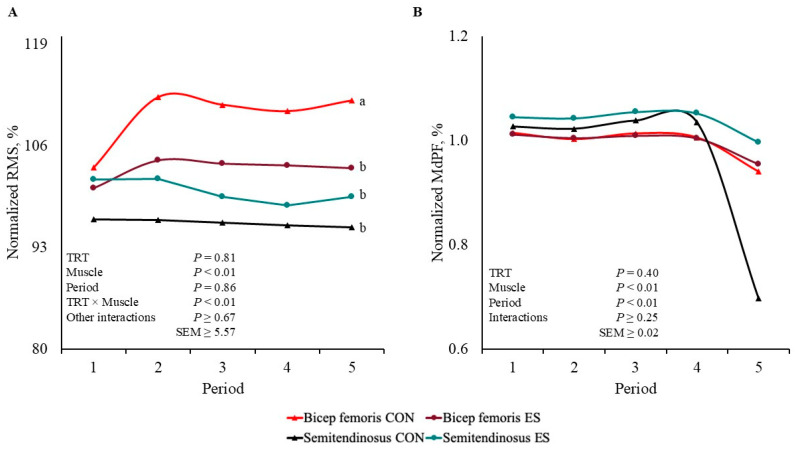
Bicep femoris and semitendinosus electrical signals normalized to the first 30 s of stimulation values of barrows subjected to an electrical stunner super-contraction (**ES**) or not (**CON**). Electromyography values were processed according to (**A**) root mean square (**RMS**) and (**B**) median power frequency (**MdPF**). ^a,b^ Lines within a panel with different subscripts differ (*p* ≤ 0.05).

**Figure 3 metabolites-15-00054-f003:**
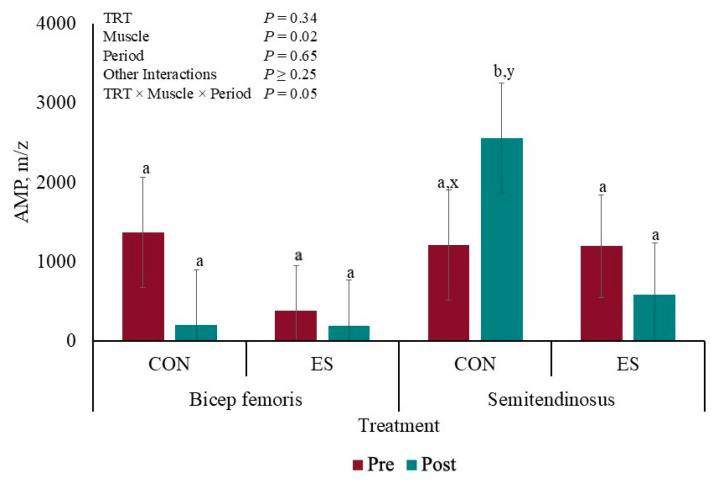
Liquid chromatography–mass spectrometry redox panel of AMP before and after muscle electrical stimulation of barrows subjected to an electrical stunner super-contraction (**ES**) or not (**CON**). ^a,b^ Means with different subscripts differ (*p* ≤ 0.05). ^x,y^ Means with different subscripts tended to differ (0.05 < *p* < 0.10).

**Figure 4 metabolites-15-00054-f004:**
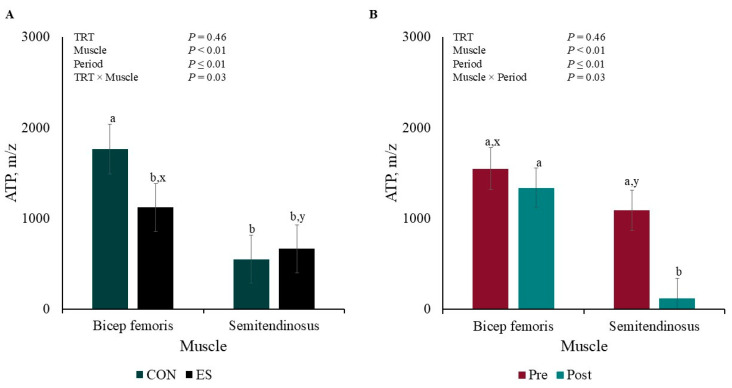
Pre- and post-muscle electrical stimulation adenosine triphosphate (**ATP**; **A**) Treatment × Muscle and (**B**) Muscle × Period interactions of barrows subjected to an electrical stunner super-contraction (**ES**) or not (**CON**). ^a,b^ Means within a panel with different subscripts differ (*p* ≤ 0.05). ^x,y^ Means within a panel with different subscripts tended to differ (0.05 < *p* < 0.10).

**Figure 5 metabolites-15-00054-f005:**
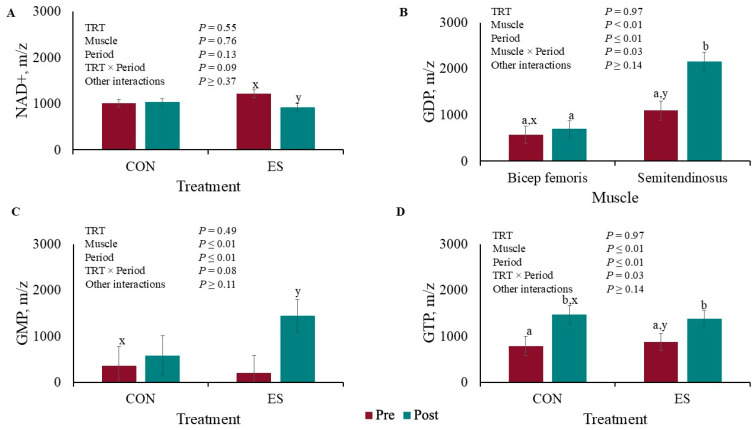
Liquid chromatography–mass spectrometry redox panel of (**A**) NAD+, (**B**) GDP, (**C**) GMP, and (**D**) GTP before and after muscle electrical stimulation of barrows subjected to an electrical stunner super-contraction (**ES**) or not (**CON).**
^a,b,c,d^ Means within a panel with different subscripts differ (*p* ≤ 0.05). ^x,y^ Means within a panel with different subscripts tended to differ (0.05 < *p* < 0.10).

**Table 1 metabolites-15-00054-t001:** Liquid chromatography–mass spectrometry redox panel.

	Treatment ^1^						*p*-Value ^2^		
	Control		Stimulated		SEM				
Muscle ^3^	Pre	Post	Pre	Post		TRT	Muscle	Period	Interactions
Bicep femoris, *m*/*z*									
ADP	1171.84	979.66	1094.07	1151.48	182.84	0.84	0.55	0.06	≥0.21
NADH	858.68	1289.03	1231.12	1395.19	539.20	0.36	0.62	0.31	≥0.34
Semitendinosus, *m*/*z*									
ADP	1276.36	1020.67	1357.12	976.57	162.08	0.84	0.55	0.06	≥0.21
NADH	813.99	686.34	926.84	1734.59	471.44	0.36	0.62	0.31	≥0.34

^1^ Barrows either being subjected to super-contraction (Stimulated) or not (Control); ^2^ TRT, Treatment; ^3^ Muscle biopsies collected before and after electrical stimulation and electromyography.

## Data Availability

The raw data supporting the conclusions of this article will be made available by the authors, without undue reservation.

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
