# Peer review of "Development of an Alternative Protocol to Study Muscle Fatigue"

_metabolites, 2025, doi:10.3390/metabo15010054_

Round 1
Reviewer 1 Report
Comments and Suggestions for Authors
The article "Development of an alternative protocol to study muscle fatigue" offers valuable insights into using EMS-induced super-contractions and EMG to replicate and assess muscle fatigue mechanisms, which could have significant implications for understanding fatigue in clinical, agricultural, and sports contexts. However, several aspects need refinement to enhance its scientific rigor and utility. The reliability concerns of EMG data, such as signal interference and flawed recordings, should be addressed by incorporating advanced noise reduction techniques or alternative methodologies. Additionally, discrepancies between RMS and MdPF findings warrant further investigation to ensure accurate replication of physiological fatigue responses. The discussion on ATP metabolism and purine derivatives could benefit from more detailed analysis of fiber-type-specific responses and their implications. Clarifying the NAD+/NADH dynamics and their alignment with known glycolytic pathways would also strengthen the study's conclusions. Addressing these points will enhance the article's impact and provide the audience with a more robust framework for understanding and applying this alternative protocol.

Author Response
Please see the attached file for our comments.
Thank you,
JMG

Reviewer 2 Report
Comments and Suggestions for Authors
The study is dedicated to development of an alternative protocol to study muscle fatigue using post-mortem barrow muscles. The novel protocol could be a promising approach to investigate muscle metabolism, however, the description of this method should be improved.
First of all, the analysis of causes of muscle fatigue in the introduction section is insufficient. The authors should briefly mention the main types of muscle fatigue (post-exercise, disuse-induced, disease-induced fatigue) and keep in mind that fatigue can be of both nervous and muscular origin. And among the metabolites connected to muscle fatigue in should be mentioned at least creatine phosphate, lactate and glycogen depletion and ROS accumulation in muscle tissue. The authors indicate that they focus on the problem of pig fatigue, but the rest of the article is dedicated only to methodological aspect of the fatigue detection protocol, but not to the analysis of the problem stated. The information of the pig fatigue problem should either be removed from the introduction or added to the discussion and conclusions.
Please, re-write the section “data collection”, as the procedure of muscle contraction initiation is uclear. At first, it is written that the animals underwent super-contraction cycle consisted of two 9 s electrical super-contraction at 570 V and 0.5-2.0 A, than the biopsies were taken (and frozen in liquid nitrogen), and the next paragraph is dedicated to “muscle contraction was induced with EMS iStim EV-805 4CH Digital TENS/EMS (Everyway Medical Instruments Co., New Taipei City, Taiwan) set at 70 Hz pulse frequency and 4:4 duty cycle (4 102 sec on, 4 sec off) for 20 min. Were there taken the two series of biopsies before and after EMS procedure? Did you perform EMS on the same muscle that had just underwent biopsy, or was it the contralateral muscle? Was it also the biopsy taken before the aforementioned super-contraction cycle?
Please provide more details about the EMS procedure. How did you ensure that all myofibers are recruited during contraction? What was the time period between super contraction cycle and animal euthanasia and between animal euthanasia and EMS? What was the method of animal euthanasia? Were the animals exsanguinated before or after the EMS? How did you ensure that post-mortem muscles had enough oxygen and glucose suppletion during EMS? Did the animal still have blood circulation during EMS?
Data were normalized to the first 30 s of each individual barrow stimulation, separately – why did you use this normalization?
Figure legends: Please specify the meaning for each individual letter, and not for groups of letters. For example, (lines 169-170) a,b Means within a panel with different subscripts di fer (P £ 0.05). x,y Means within a panel with different subscripts tended to differ (0.10 < P 170 > 0.05) – what is the difference between a and b or between x and y?
Author Response
Please see the attached file for our responses.
Thank you,
JMG

Reviewer 3 Report
Comments and Suggestions for Authors
The manuscript needs revision. Please refer to comments given in the text of reviewed attached file of the manuscript.

Author Response

(The authors gave the same response as above.)

Round 2
Reviewer 2 Report
Comments and Suggestions for Authors
The manuscript have been rather improved after revision. However, the sample of raw EMG for each experimental group and for each analysed time-point should be added either to article figures or to supplementary materials to improve data presentation.
Author Response
Thank you for the comments and helping make the manuscript better. I hope we interpreted your question correctly and we have added a new "Figure 1" to satisfy this request.
Thank you,
JMG